# Health extension service utilization and associated factors in East Gojjam zone, Northwest Ethiopia: A community-based cross-sectional study

**Bewket Yeserah Aynalem**\*, **Misganaw Fikrie Melesse**

Department of Midwifery, Debre Markos University, Debre Markos, Ethiopia

\* by123bewket@gmail.com

## Abstract

### Introduction

Health Extension Program is a preventive, promotive, and basic curative service targeting households to improve the health status of families with the effective implementation of 16 health extension packages. We, therefore, did this study to assess health extension package utilization and associated factors in the East Gojjam zone, Northwest Ethiopia.

### Methods

A community-based mixed cross-sectional study was conducted on households of East Gojjam Zone, from January 1 to April 30, 2020. A multistage sampling procedure was used to select 806 study participants in this study. We used EPI info version 7 for data entry and SPSS version 24 software for cleaning and analysis. Variables having a P-value of less than 0.25 in the bivariate logistic regression analysis were fitted into the multivariable logistic regression model. The 95% confidence interval of odds ratio was computed and a variable having P-value less than 0.05 in the multivariable logistic regression analysis was considered as statistically significant.

### Results

The study showed that 119 (14.8%) respondents have utilized health extension packages. Knowledge health extension package (AOR = 1.84, 95% CI: 1.22, 2.79), residence (AOR = 3.55, 95% CI: 1.99,6.33),visited health post(AOR = 1.63, 95% CI: 1.054,2.50), home visited by health extension worker (AOR = 1,68, 95% CI: 1.025,2.74) and involving in model family training(AOR = 2.10, 95% CI: 1.38,3.215) were significant factors for health extension service utilization.

### Conclusion

The magnitude of health extension service utilization was low since the Ethiopian government recommends 100% health extension service utilization coverage. Knowledge of health

---

**Data Availability Statement:** All relevant data are within the manuscript and its Supporting information files.

**Funding:** The authors received no external funding for this work.

**Competing interests:** The authors have declared that no competing interests exist.

extension package, residence, health post-visit, home visit, and model family training were significant factors for health extension service utilization. So expanding the model family training and strict home-to-home visit especially in rural areas may increase the health extension package utilization.

## Introduction

Establishing an effective and responsive health service delivery system is an integral part of the overall development that aims to reduce poverty and achieve economic growth and development [1, 2]. The Health Extension Program (HEP) is a package of preventive, promotive, and basic curative services targeting households to improve the health status of families with their full participation [3]. HEPs include disease prevention and control, family health service, hygiene, and environmental sanitation, health education, and communication [1]. The former Frontline Health workers or traditional birth attendants (TBAs) are incorporated into the system by serving as volunteers that work under the supervision of the health extension worker (HEW) [4]. Thus, HEP will serve as a primary vehicle for prevention, health promotion, behavioral change communication, and basic curative care [5]. Globally many countries, especially developing countries are striving to achieve universal health coverage starting from the "Health for All" movement of 1978 by the World Health Organization (WHO) or the Almata declaration [6]. The Government of Ethiopia has implemented the Health Extension package (HEP) since 2003 to improve primary health coverage at the grassroots level [7] and also agreed to this movement and tried to address universal health coverage mainly primary health care (PHC) throughout the country [8].

The center for the HEP activity is a health post (HP) located in each smallest administrative unit of the country and is staffed by two female Health Extension Workers (HEWs) who receive one year's training and a regular salary from the government [9, 10]. HEP was published to have public sector facilities provide a minimum standard of care that fosters an integrated service delivery approach [11]. And it delivers cost-effective basic services to all Ethiopians, mainly women, and children [12]. More than 75% of the health problem in Ethiopia is largely attributed to preventable communicable diseases and under-nutrition. The prevalence of these diseases is mainly due to poor socio-economic conditions, a low level of awareness about health, and inadequate health service delivery across the country [13]. Concerning this, the Ethiopian government designed a sound able plan to achieve universal access to primary health care by preparing a health sector development program (HSDP). This plan was aimed to address the service coverage problem of the health system through an accelerated expansion and strengthening of primary health care services [1, 12].

There are some factors associated with health extension service utilization in the previous study: these are knowledge of community on health extension service, age, occupation, community participation in the planning of health extension activities, and graduation of model family [18].

According to the Ethiopian Demographic and Health Survey of 2016, 38% of children under the age of five years suffered from stunting due to malnutrition; the Amhara region accounts for the highest proportion (46%) and the prevalence of water-borne and water-washed diseases were high. The Mini EDHS 2019 report also indicates under 5 mortality 30%, maternal mortality 412 per100,000 live births, and unmet need for family planning is 22% [14]. Therefore, this study aims to assess HEP utilization and its factors in East Gojjam Zone.

## Methods

### Study areas and period

East Gojjam zone is located in the Amhara region 300 km from Addis Ababa, a capital city of Ethiopia, and 265 km from Bihar Dar, the capital city of Amhara. It is bordered on the south by the Oromia Region, on the west by West Gojjam, on the north by South Gondar, and on the east by South Wollo. As the zonal health office report showed East Gojjam zone has a total population of 2,719,118 and 632,353 households. East Gojjam zone has also 21 Woreda, 480 Kebeles, 423 health posts, 102 health centers, 9 primary hospitals, and one referral hospital. The study was conducted from January 1, 2020, to April 30, 2020.

### Study design

A community-based cross-sectional study design was conducted.

### Study participants

The source population was all households who live in the East Gojjam zone. The study population was households in the east Gojjam zone during the study period in the selected Kebeles. Study participants age 18 and above who reside above six months in the study area were included and participants who were mentally ill during the data collection period were excluded.

### Sample size determination

The sample size was determined based on a single population proportion formula assumption. The expected proportion of HEP utilization (39%) from the previous study in Ethiopia Abuna Gindeberet, West Shoa Zone [15], and a 5% confidence limit (margin of error) was used.

$$initial sample size = \left(Z\frac{a}{2}\right)^2 * \frac{p(1-p)}{w2} = 1.96^2 * \frac{0..39(1-0.39)}{(0.05)^2} = 366$$

Considering design effect 2 since it had two stages and the sample size was calculated as $366^*2 = 732$. Then the non-response rate was also considered to be 10% and $732^*0.10 = 74$. Then the final sample size was 732+74 = 806.

### Sampling techniques

A multistage sampling technique was used and firstly all the woreda found in the east Gojjam zone were listed in a frame. Then five out of the 21 woredas were selected by the lottery method. Again 4 kebeles, from Bibugn woreda, 5 kebeles from Debre Elias woreda, 5 kebeles from Dejen woreda, and 5 kebeles from Andede woreda were selected with the lottery method. The size of households consisting of the eligible population to be selected from each kebele was determined proportionally based on the size of the study units and the $k^{th}$ value was computed for each selected kebele. Any single individual age 18 and above of the selected household was interviewed. In the case of absenteeism, after three repeated visits the next eligible household was included in the study.

### Study variables

**Dependent variable.** Health extension service utilization.

**Independent variables.** Age, marital status, religion, educational status, occupation, religion sex, residence, family size, knowledge, HP distance from home, HP visit, home visited by HEWs, model family training, model family graduation.

## Operational definitions

**Health extension service utilization.** Anyone who implements at least 75% of the national health extension program packages [16].

**Satisfactory knowledge.** Respondents who scored 75% and above of the HEPs knowledge questions [15].

**Model households.** Households that attended at least 75% of the training and implemented at least 75% of the HEP and given certificates of completion [10].

**Health extension package.** Is a package of preventive, promotive, and basic curative services targeting households to improve the health status of families with their full participation [3].

## Data collection and data quality control

To assure the data quality, data were collected with face-to-face interviews by three trained diploma nurses after two-day data collection training was given to them together with three BSc holder supervisors. The questionnaire was structured and pre-tested which was first prepared in English and translated to local (Amharic) language and then again translated back to English. A pretest was conducted on 40 households of the sample size other than the study area and the necessary correction on the tool was employed accordingly.

## Data processing and analysis

Epi Info version 7 software was used for data entry and SPSS version 24 for used for analysis. Bivariate logistic regression was employed to identify an association between independent and dependent variables. Variables having a P-value of less than 0.25 in the bivariate logistic regression analysis were fitted into the multivariable logistic regression model. The 95% confidence interval of odds ratio was computed and a variable having P-value less than 0.05 in the multivariable logistic regression analysis was considered as statistically significant.

## Ethical clearance

Ethical clearance was obtained from the research committee (Institutional Research Ethics Review Committee) of Debre Markos University with ethical approval number HSC/R/C/Ser/ Co/341/06/12 and was submitted to the East Gojjam zone health bureau. Ethical clearance and formal letters were also obtained from the Debre Markos University and were submitted to the East Gojjam zone health bureau and permission was obtained. Finally, written informed consent was also obtained from each study participant in the sampled households.

## Results

### Socio-demographic characteristics

All 806 study participants responded to the questionnaire, giving a response rate of 100%. All of the participants were Amhara in ethnicity and 758 (94.1%) of the participants are orthodox Christian. Five hundred thirteen (63.6%) were living in a rural area with agriculture as a source of income (Table 1).

**Table 1. Sociodemographic characteristics of respondents (n = 806) in East Gojjam Zone, Northwest Ethiopia, 2019.**

| Variable | Frequency | Percent |
|---|---|---|
| **Age**(in years) | | |
| 18–24 | 30 | 3.7 |
| 25–39 | 357 | 44.3 |
| ≥40 | 419 | 52 |
| **Marital status** | | |
| Single | 48 | 5.9 |
| Married | 696 | 86.4 |
| Widowed | 20 | 2.5 |
| Divorced | 42 | 5.2 |
| **Religion** | | |
| Orthodox | 758 | 94.1 |
| Muslim | 38 | 4.7 |
| Protestant | 10 | 1.2 |
| **Educational status** | | |
| No formal education | 485 | 60.2 |
| Primary education | 126 | 15.6 |
| Secondary education | 90 | 11.2 |
| College and above | 105 | 13 |
| **Occupation** | | |
| Housewife | 46 | 5.7 |
| Self-employee (doing own small business) | 138 | 17.6 |
| Private employee(salaried in the nongovernmental sector) | 44 | 5.5 |
| Government employee | 65 | 8.1 |
| Farmer | 513 | 63.6 |
| **Residence** | | |
| Rural | 513 | 63.6 |
| Urban | 293 | 36.4 |
| **Source of income** | | |
| Agriculture | 513 | 63.6 |
| Other* | 293 | 36.4 |
| **Sex** | | |
| Male | 152 | 18.9 |
| Female | 654 | 81.1 |
| **Family size** | | |
| 1–4 | 316 | 39.2 |
| 4$^{+}$ | 490 | 60.8 |

## Service-related characteristics

Less than half (37%) of the study participants have transport access to the health post. Two hundred-one (90.5%) of the participants got a good approach from HEWs. Again around 208 (93.7%) participants got participants needed (Table 2).

## Knowledge and modeling related characters

Among the respondents, only 311(38.6%) had satisfactory knowledge of HEPs. About 259 (32.1%) of respondents have participated in model family training of HEPs. Of these 249

**Table 2. Service related issues among respondents (n = 806) in East Gojjam Zone, Northwest Ethiopia, 2019.**

| Variable | Frequency | Percent |
|---|---|---|
| Transport access to the health post | | |
| Yes | 298 | 37 |
| No | 508 | 63 |
| Visited health post | | |
| Yes | 222 | 27.5 |
| No | 584 | 72.5 |
| Reasons for visiting health post | | |
| For antenatal care | 116 | 48.7 |
| For family planning | 91 | 38.2 |
| For other services | 31 | 13.1 |
| health extension worker approach | | |
| Good | 201 | 90.5 |
| Bad | 21 | 9.5 |
| Returned without getting the service needed | | |
| Yes | 14 | 6.3 |
| No | 208 | 93.7 |
| Reasons returned without getting the service | | |
| Absence of health extension workers | 8 | 57.1 |
| Unavailability of the service | 6 | 42.9 |
| Home visited by a health extension worker | | |
| Yes | 544 | 67.5 |
| No | 262 | 32.5 |

(96.1%) had been graduated. One hundred ten (44.2%) of these had 1–2 years, 122(49%) had 3 years and 17(6.8%) had greater than 4 years after model family graduation.

## Utilization of the health extension package and its associated factors

One hundred nineteen (14.8%) of the study population have utilized health extension packages [95% CI: 12.4, 17.2] (Table 3). Bivariate logistic regression was used to identify an association between independent and outcome variables. Variables with P-values <0.25 in binary logistic regression (educational status, residence, knowledge, health post-visit, home visited by HEW, and model family training) continued to be fitted into the multivariable logistic regression.

After controlling the effect of other variables with multivariable logistic regression analysis, residence [AOR: 3.549(95% CI: 1.99, 6.33)], knowledge [AOR: 1.84(95% CI: 1.216, 2.796)] health post-visit [AOR: 1.63(95% CI: 1.054, 2.50)], home visited by HEW [AOR: 1.68(95% CI: 1.03,2.740)] and Graduated from model family training [AOR: 2.10(95% CI: 1.38,3.22)] were significant factors for health extension package utilization (Table 4).

## Discussion

The study finding of health extension package utilization (14.8%) in the current study was higher than the study done in Hosanna, Hadya Zone, Southern Ethiopia (2.352%) [17]. The possible explanation for this difference might be the difference in socio-economic status and cultural aspects and differences in the level of awareness about health extension packages.

But the result of this study is lower than the study conducted in Abuna Gindeberet district, West Shoa Zone, Ethiopia (39%) [18]. The possible reasons for this difference could be the

**Table 3.** Utilization of health extension package (n = 806) in East Gojjam Zone, Northwest Ethiopia, 2019.

| Package | Frequency | percent |
|---|---|---|
| **A: Hygiene and sanitation** | | |
| Proper and safe excreta disposal | | |
| Yes | 366 | 45.4 |
| No | 440 | 54.6 |
| Proper and safe solid and liquid management | | |
| Yes | 268 | 33.3 |
| No | 538 | 66.7 |
| Personal hygiene | | |
| Yes | 310 | 38.5 |
| No | 496 | 61.5 |
| Water supply and hygiene | | |
| Yes | 142 | 17.6 |
| No | 664 | 82.4 |
| Proper and safe home environment | | |
| Yes | 469 | 58.2 |
| No | 337 | 41.8 |
| Insects and rodents control | | |
| Yes | 592 | 73.4 |
| No | 214 | 26.6 |
| Food hygiene | | |
| Yes | 471 | 58.4 |
| No | 335 | 41.6 |
| **B: Communicable disease prevention and control** | | |
| HIV/AIDS and TB prevention and control | | |
| Yes | 610 | 75.7 |
| No | 196 | 24.3 |
| Malaria prevention and control | | |
| Yes | 461 | 57.2 |
| No | 345 | 42.8 |
| First aid | | |
| Yes | 77 | 9.6 |
| No | 729 | 90.4 |
| Youth reproductive health care | | |
| Yes | 179 | 22.2 |
| No | 627 | 77.8 |
| Child and maternal health care | | |
| Yes | 481 | 59.7 |
| No | 325 | 40.3 |
| Maternal and child nutrition | | |
| Yes | 387 | 48 |
| No | 419 | 52 |
| Immunization | | |
| Yes | 583 | 72.3 |
| No | 223 | 27.7 |
| Family planning | | |
| Yes | 518 | 64.3 |
| No | 288 | 35.7 |

(*Continued*)

**Table 3.** (Continued)

| Package | Frequency | percent |
|---|---|---|
| C: **Communication** | | |
| Communication and health education | | |
| Yes | 577 | 71.6 |
| No | 229 | 28.4 |
| **Over all utilization** | | |
| Yes | 119 | 14.8 |
| No | 687 | 85.2 |

difference in study area coverage. This study was done at a zonal level that covers a wide area that might have great variation in health extension package utilization across the zone and leads to low coverage of HEP utilization. Again this finding is lower than the study done in Gulelle sub-city Administration, Addis Ababa, Ethiopia (86%) [15]. The possible explanation for this difference can be the study setting, socio-economic status of the community, and living standards. Our study was conducted in areas including the rural community which is unthinkable to be comparable with a community living in Addis Ababa. People in the Gulelle sub-city administration utilize the health extension packages on their own since they understand the importance of these packages whereas the rural community considers the importance of these packages is not for them rather either for the health extension workers or for the government. The other possible reason may be a limitation in basic infrastructures in rural communities relative to the Gulelle sub-city administration community.

As shown in this study, residence was one of the significant factors for the utilization of health extension package utilization. Urban dwellers were 3.55 times more likely to utilize the health extension packages as compared with rural households (AOR = 3.55, 95% CI: 1.99, 6.33). This finding was supported by the studies done in Kombolcha town district, East Hararghe Zone, Eastern Ethiopia, and Esera district of Southern Ethiopia [19, 20]. The possible explanation for this might be urban residents might have higher information access about health extension packages than the rural dwellers and there might have infrastructure assess to utilize these package.

Participants with a satisfactory knowledge of health extension packages were 1.84 times more likely to utilize these packages than respondents with their counterparts (AOR = 1.84, 95% CI: 1.22, 2.79). This result is supported by other studies conducted in Ethiopia as nationwide, Abuna Gindeberet District, West Shoa Zone, Ethiopia, and Hadya Zone, Southern Ethiopia [17, 18, 21, 22]. knowledge is the entry point for any activity, so respondents who know the health extension package could utilize those packages than their counterparts. Again participants who got the opportunity to know about health extension packages may have an effort to utilize it than those who do not have.

The other variable which was significantly associated with the health extension service utilization was health post-visit. Respondents who had visited the health post were 1.63 times more likely to utilize the health extension packages than those who did not visit the health post (AOR = 1.63, 95% CI: 1.054, 2.50). This finding was supported by a nationwide finding of Ethiopia [23]. The possible explanation for this difference might be as the respondents visit the HP they may have the opportunity to be familiar with the HEWs and get enough information about the HEP utilization than those who did not visit. When the respondents visit the HP, they can access the HEWs to talk frankly about HEP utilization with the aid of demonstration

**Table 4. Bivariable and multivariable analysis of factors associated with utilization of health extension package in East Gojjam Zone, Northwest Ethiopia, 2019.**

| Variable | Utilized | | Crude OR[95%CI] | AOR[95%CI] |
|---|---|---|---|---|
| | **No** | **Yes** | | |
| **Age** | | | | |
| 18–24 | 26 | 4 | 1 | |
| 25–39 | 307 | 50 | 0.94 (.35, 3.16) | |
| ≥40 | 354 | 65 | 1.194(.403,3.53) | |
| **Marital status** | | | | |
| Single | 41 | 7 | 1 | |
| Married | 590 | 106 | 1.50 (.46,2.41) | |
| Widowed | 18 | 2 | .651(.12, 3.44) | |
| Divorced | 38 | 4 | .617(.17,2.27) | |
| **Religion** | | | | |
| Orthodox | 644 | 114 | 1 | |
| Muslim | 35 | 3 | .48 (.15,1.60) | |
| Protestant | 8 | 2 | 1.412(.29, 6.74) | |
| **Educational status** | | | | |
| No formal education | 434 | 51 | **1** | 1 |
| Primary education | 99 | 27 | **3.32 (1.39,3.88)** | 1.126(.60,2.09) |
| Secondary education | 74 | 16 | **1.84 (.99, 3.39)** | .64 (.302,1.39) |
| College and above | 80 | 25 | **2.66(1.56,45)** | .774(.370,1.615) |
| **Sex** | | | | |
| Female | 564 | 90 | 1 | |
| Male | 123 | 29 | 1.48(.93, 2.35) | |
| **Residence** | | | | |
| Rural | 466 | 47 | **1** | **1** |
| Urban | 221 | 72 | **3.23 (2.16, 4.82)** | **3.55(1.99,6.33)** ** |
| **Family size** | | | | |
| 1–4 | 269 | 47 | 1 | |
| Above 4 | 418 | 72 | .99(.66,1.47) | |
| **Knowledge** | | | | |
| Unsatisfactory | 437 | 58 | **1** | **1** |
| Satisfactory | 250 | 61 | **1.84(1.24, 2.72)** | **1.84 (1.22, 2.79)** ** |
| **Visited health post** | | | | |
| No | 511 | 73 | **1** | **1** |
| Yes | 176 | 46 | **1.83 (1.22, 2.75)** | **1.63(1.05, 2.50)** |
| **Home visited by HEW** | | | | |
| No | 237 | 25 | **1** | **1** |
| Yes | 450 | 94 | **1.98(1.24,316)** | **1.68(1.03, 2.74)** |
| **Participated in model family training** | | | | |
| No | 484 | 63 | **1** | **1** |
| Yes | 203 | 56 | **2.12(1.43,3.15)** | **2.10 (1.38, 3.22)** |
| **Graduated from model family training** | | | | |
| No | 9 | 1 | 1 | |
| Yes | 194 | 55 | 2.55 (.32, 20.56) | |

** P value< 0.001, 1reference,

*housewife self-employee, private employee, and government employee.

in addition to the home to home visit than those who do not visit. Visiting the HP by itself is using the HEP like ANC follow-up, HIV/AIDS counseling, family planning access, malaria prophylaxis, and prevention access, getting health education, and other packages. Therefore, respondents who visit HP may easily utilize HEP than those who did not.

Respondents from the household which is visited by the HEW were 1.676 times more likely to utilize the HEP than from not visited households (AOR = 1.68, 95% CI: 1.03, 2.74). This result was supported by a study done in Akaki district, Addis Ababa, and nationwide studies conducted in Ethiopia [23–25]. Visiting the households by the HEWs is the key factor for HEP utilization by the community. During the HEWs home to home visit, they perform lots of activities concerning HEP like registering pregnant women with their expected date of delivery, proper utilization of latrine, kitchen, insect acid-treated net, family planning, diagnose ill individuals in the household, and refer to health institution, provide health education and proper disposal of waste materials. So study participants whose houses are visited by the HEWs can utilize HEP than which is not visited.

Participants who had been involved in model family training were 2.10 times more likely to implement the health extension packages than those who had not participated (AOR = 2.10, 95% CI:1.377, 3.22). This result was supported by a study conducted in Nefas Silk Lafto Sub-city, Addis Ababa, Ethiopia [26]. The possible explanation might be involving the model family training regarding the health extension package increases awareness about these packages utilization than those who did not participate. The other reason could be when participants are involved in the model family training; they may have inspired to utilize the HEP than those who do not participate.

## Conclusion

The magnitude of health extension service utilization was low as compared to other studies. Knowledge, residence, health post-visit, home visit, and model family training were significant factors for health extension service utilization. So expanding the model family training and strict home-to-home visit especially in rural areas may increase the health extension package utilization.

## Limitation and strength

The strength of this study was the data source which was primary data collected directly from the community that makes it more accurate and representative for the study population. Difficulty of data collection from the community due to COVID-19 in terms of cost (availing sanitizer, face masks and transport) and the nature of study design (cross sectional study design) which cannot show the cause and effect relationship between the independent and the outcome variables were the limitation of this study.

## Supporting information

**S1 Dataset.**
(SAV)

**S1 Questionnaire.**
(DOCX)

## Acknowledgments

We would like to thank Debremarkos University for permitting this research and we gratefully acknowledge all study individuals for their participation in the study.

## Author Contributions

**Conceptualization:** Bewket Yeserah Aynalem, Misganaw Fikrie Melesse.

**Data curation:** Bewket Yeserah Aynalem.

**Formal analysis:** Bewket Yeserah Aynalem, Misganaw Fikrie Melesse.

**Funding acquisition:** Bewket Yeserah Aynalem.

**Investigation:** Bewket Yeserah Aynalem.

**Methodology:** Bewket Yeserah Aynalem.

**Project administration:** Bewket Yeserah Aynalem.

**Resources:** Bewket Yeserah Aynalem.

**Software:** Bewket Yeserah Aynalem, Misganaw Fikrie Melesse.

**Supervision:** Misganaw Fikrie Melesse.

**Validation:** Misganaw Fikrie Melesse.

**Visualization:** Bewket Yeserah Aynalem.

**Writing – original draft:** Bewket Yeserah Aynalem.

**Writing – review & editing:** Misganaw Fikrie Melesse.

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
