## [Decision Letter · Decision Letter 0]

15 Apr 2021

PONE-D-20-30163

Health extension service utilization and associated factors in east Gojjam zone, North West Ethiopia: community based cross-sectional study

PLOS ONE

Dear Dr. Aynalem,

Thank you for submitting your manuscript to PLOS ONE. After careful consideration, we feel that it has merit but does not fully meet PLOS ONE’s publication criteria as it currently stands. Therefore, we invite you to submit a revised version of the manuscript that addresses the points raised during the review process.

The manuscript has been evaluated by two reviewers, and their comments are available below.

The reviewers have raised a number of concerns that need attention. They have requested additional references and citations to the current literature in support of the conceptual framework used to guide the design of the research questions. Furthermore the reviewers have expressed overlapping needs for further clarifications on the study design as well additional information on methodological aspects of the study (such as the justification for the sample size calculations and revision to the statistical analysis). Finally the article must be presented in an intelligible fashion and written in clear, correct, and unambiguous English, (https://journals.plos.org/plosone/s/criteria-for-publication#loc-5 ). Please note that PLOS ONE cannot provide copyediting for manuscripts and, as a result, we advice that the manuscript is copy edited by a native English speaker.

Could you please revise the manuscript to carefully address the concerns raised?

We look forward to receiving your revised manuscript.

Kind regards,

Lucinda Shen, MSc

Staff Editor

PLOS ONE

Journal Requirements:

- https://journals.plos.org/plosone/article?id=10.1371%2Fjournal.pone.0231307

The text that needs to be addressed involves page 5.

In your revision ensure you cite all your sources (including your own works), and quote or rephrase any duplicated text outside the methods section.

Further consideration is dependent on these concerns being addressed.

3. Thank you for stating the following after the Acknowledgments Section of your manuscript:

'Funding

Fund was obtained from Debremarkos University'

a. Please remove any funding-related text from the manuscript and let us know how you would like to update your Funding Statement. Currently, your Funding Statement reads as follows: 'no'

Please clarify the sources of funding (financial or material support) for your study. List the grants or organizations that supported your study, including funding received from your institution.State what role the funders took in the study. If the funders had no role in your study, please state: “The funders had no role in study design, data collection and analysis, decision to publish, or preparation of the manuscript.”If any authors received a salary from any of your funders, please state which authors and which funders.

*Please include your amended statements within your cover letter; we will change the online submission form on your behalf.*

Reviewers' comments:

Reviewer's Responses to Questions

**Comments to the Author**

1. Is the manuscript technically sound, and do the data support the conclusions?

Reviewer #1: Yes

Reviewer #2: Partly

2. Has the statistical analysis been performed appropriately and rigorously? 

Reviewer #1: Yes

Reviewer #2: Yes

3. Have the authors made all data underlying the findings in their manuscript fully available?

Reviewer #1: Yes

Reviewer #2: Yes

4. Is the manuscript presented in an intelligible fashion and written in standard English?

Reviewer #1: Yes

Reviewer #2: No

5. Review Comments to the Author

Reviewer #1: Reviewer comment to author’s

Title: - Health extension service utilization and associated factors in east Gojjam zone, North

West Ethiopia: community based cross-sectional study.

Thank you very much editor for give me a chance to review this interesting manuscript. I thank you for authors for your effort and interesting work. I think this work adds value for the scientific community and the community in which the study done after the following issues addressed. English language editing will benefit for improvement.

General comment

Before submitted any manuscript to journal you should give line number and follow the guideline of the journal. This manuscript does not have line number and it is difficult to give line-by-line comments. Anyways my comment given each session of the manuscript find it and address it.

Abstract

Introduction

The introduction section of the abstract does not show the gap. What is the reason to study this research? It needs rewriting explain the gap and reason to conduct this study needed. Before you talk about the significance of the study, you should tell the gap.

Method

The method session needs some modification. The model fitness and include the bivariable analysis variable selection p-value

Result

The abbreviation HEW did not explained before but written their it not write. The abstract session should be free and abbreviation

Conclusion: - In the first sentence, you said, “The magnitude of health extension service utilization was low.” What is your cutting point to say high/low? Such types of conclusion is blind and not informative.

Body of the manuscript

Introduction

In your introduction session, I did not see any factors associated with health extension service utilization what previous scholars explore. It needs one paragraph the reviews the associated factors of health extension service utilization in Ethiopia. Include it in your next revision part.

Method

In the study design session said” Community based mixed cross-sectional study design” what does it mean? I did not see any qualitative part. I need a justification such types of study design.

What your justification in your study participants session you included participants in age >=18 years?.

In your sample size calculation formula you only calculated for prevalence of health extension service utilization. Why not calculate for factors. You are expected to calculate for both the magnitude and the factor then you will take sample size which large. Why that not done?

In your sampling technique from 5/21 woredas why only 5? Justify it. Again the seleceted kebeles from total woredas ? how determine the number of kebele for each woreda? Justfy it.

In your data processing and analysis you used 0.25 as a variable screening but there is no any justification. Justify why that?

The model fitness issue is not presented. Why? Justify it. Include in the revision part

Did you check multicolleanity among explanatory variables? if not why justify it.

Did you check assumptions of binary logistic regression model?

What variable selection method you used? Forward/backward/stepwise?

Result

In the result section it said that “giving a response rate of 100%.” How it could be? Unbelievable result. justify it.

In the “Knowledge and modeling related characters” has no table citation? Why?

In the model interpretation session you said that “After controlling the effect of other variables

with binary logistic regression, educational status, residence, knowledge, health postvisit,

home visited by HEW, and model family training continued to be significantly associated with utilization of HEPs (P-values<0.25).” how other variables effect controlled in the bivariable analysis part? Justify it.

In the final model result interpretation section “After controlling the effect of other variables with multivariable logistic regressionanalysis, residence [AOR: 3.549(95% CI: 1.99, 6.328)], knowledge [AOR: 1.844(95%CI: 1.216, 2.796)] health post-visit [AOR: 1.625(95% CI: 1.054, 2.504)], home visited by HEW [AOR: 1.676(95% CI: 1.025,2.740)] and Graduated from model family training [AOR: 2.104(95% CI: 1.377,3.215)] were significant factors for health extension

package utilization (Table 4 ).”

Such type of writing should be in the abstract session, you are expected interpret the result explicitly. Correct it during the revision.

Discussion

I found your result and discussion section mixed. You should separate them. If the result is correctly, interpret in the result session no need to repeat it in the discussion session.

In the discussion section your speculation needs citation example “The other possible reason may be a limitation in basic infrastructures in rural communities relative to the Gulelle sub-city administration community.” Need to be supported by evidence…..works for all discussion session.

I did not see any strength and limitation of the study. Why ? justify it?

Conclusion

The conclusion and recommendation should be based on your findings.

Declaration session

No Abbreviations and Acronyms

Totally you did not follow the journal guideline please follow it and correct it accordingly.

Reviewer #2: Major revision is needed in terms of addressing the comments that are indicated in the methods section, results and discussion section.

The work is essential to inform policy makers. However, clarity on the further work to address comments is needed.

Please find my review attached.

6. PLOS authors have the option to publish the peer review history of their article (what does this mean?). If published, this will include your full peer review and any attached files.

Reviewer #1: **Yes: **Zemenu T.

Reviewer #2: **Yes: **Abera Kumie

---

## [Author Response · Author response to Decision Letter 0]

13 May 2021

all comments are addressed, please help me to publish this paper as soon as possible.

---

## [Decision Letter · Decision Letter 1]

28 Jun 2021

PONE-D-20-30163R1

Health extension service utilization and associated factors in east Gojjam zone, North West Ethiopia: community based cross-sectional study

PLOS ONE

Dear Dr. Aynalem,

Thank you for submitting your manuscript to PLOS ONE. After careful consideration, we feel that it has merit but does not fully meet PLOS ONE’s publication criteria as it currently stands. Therefore, we invite you to submit a revised version of the manuscript that addresses the points raised during the review process.

We look forward to receiving your revised manuscript.

Kind regards,

Ammal Mokhtar Metwally, Ph.D (MD)

Academic Editor

PLOS ONE

Journal Requirements:

Reviewers' comments:

Reviewer's Responses to Questions

**Comments to the Author**

1. If the authors have adequately addressed your comments raised in a previous round of review and you feel that this manuscript is now acceptable for publication, you may indicate that here to bypass the “Comments to the Author” section, enter your conflict of interest statement in the “Confidential to Editor” section, and submit your "Accept" recommendation.

Reviewer #1: All comments have been addressed

Reviewer #3: (No Response)

Reviewer #4: All comments have been addressed

2. Is the manuscript technically sound, and do the data support the conclusions?

Reviewer #1: Yes

Reviewer #3: Yes

Reviewer #4: Yes

3. Has the statistical analysis been performed appropriately and rigorously? 

Reviewer #1: Yes

Reviewer #3: (No Response)

Reviewer #4: Yes

4. Have the authors made all data underlying the findings in their manuscript fully available?

Reviewer #1: Yes

Reviewer #3: (No Response)

Reviewer #4: Yes

5. Is the manuscript presented in an intelligible fashion and written in standard English?

Reviewer #1: Yes

Reviewer #3: Yes

Reviewer #4: Yes

6. Review Comments to the Author

Reviewer #1: Thanks authors to addressed all comments. This manuscript need to published and suitabl for publication in PLOS ONE.

Reviewer #3: I found this article very interesting, representing a well designed intervention. The report is totally satisfactory and I suggest to publish it in the present form

Reviewer #4: The authors identified factors associated to health extention service utilization in Noth West Ethipia. The research is original, the topic is relevant and of a high importance in its field. The statistical analysis been performed appropriately and rigorously. However, I have some suggestion to authors.

1. The formula for sample size calculation, W2 is the precision in the parameter estimation, it is different from the alpha (0.05) which is the margin of error (or 95% CI).

2. In the results section, page 7, the first paragraph. After controlling the effect of other variables with binary logistic regression, educational........continued to be significantly associated (p-values<0.25). there is contradiction we can't control in bivariate regression and p-values<0.25 is not sigificant. Please check this paragraph.

7. PLOS authors have the option to publish the peer review history of their article (what does this mean?). If published, this will include your full peer review and any attached files.

Reviewer #1: **Yes: **Zemenu Tadesse Tessema

Reviewer #3: **Yes: **Diego Serraino, MD

Reviewer #4: **Yes: **Hedia Bellali

---

## [Author Response · Author response to Decision Letter 1]

28 Jun 2021

Dear reviewers, I would like to forward my gratitude to you for your constructive comments. I have learnt many things from your comments.

Dear 4th reviewer, I have accepted your comments and some correction are done based on your comments.

---

## [Decision Letter · Decision Letter 2]

21 Jul 2021

PONE-D-20-30163R2

Health extension service utilization and associated factors in east Gojjam zone, North West Ethiopia: community based cross-sectional study

PLOS ONE

Dear Dr. Aynalem,

Thank you for submitting your manuscript to PLOS ONE. After careful consideration, we feel that it has merit but does not fully meet PLOS ONE’s publication criteria as it currently stands. Therefore, we invite you to submit a revised version of the manuscript that addresses the points raised during the review process.

Before accepting your manuscript, minor revision is required.

When submitting your revision, please consider addressing these requirements:

1. Please ensure including in the manuscript after the conclusion strengths and the limitation of your study

2. Please provide additional details regarding the ethical approval number  and the name of the ethical committee in the ethics statement in the Methods.

3. Add the authors contribution as per Plos1 Guidelines 

We look forward to receiving your revised manuscript.

Kind regards,

Ammal Mokhtar Metwally, Ph.D (MD)

Academic Editor

PLOS ONE

Journal Requirements:

Reviewers' comments:

Reviewer's Responses to Questions

**Comments to the Author**

1. If the authors have adequately addressed your comments raised in a previous round of review and you feel that this manuscript is now acceptable for publication, you may indicate that here to bypass the “Comments to the Author” section, enter your conflict of interest statement in the “Confidential to Editor” section, and submit your "Accept" recommendation.

Reviewer #3: (No Response)

Reviewer #4: All comments have been addressed

2. Is the manuscript technically sound, and do the data support the conclusions?

Reviewer #3: Yes

Reviewer #4: Yes

3. Has the statistical analysis been performed appropriately and rigorously? 

Reviewer #3: Yes

Reviewer #4: Yes

4. Have the authors made all data underlying the findings in their manuscript fully available?

Reviewer #3: Yes

Reviewer #4: Yes

5. Is the manuscript presented in an intelligible fashion and written in standard English?

Reviewer #3: Yes

Reviewer #4: Yes

6. Review Comments to the Author

Reviewer #3: I do not have further comments - in my opinion the paper is now ready to be published in the present form

Reviewer #4: (No Response)

7. PLOS authors have the option to publish the peer review history of their article (what does this mean?). If published, this will include your full peer review and any attached files.

Reviewer #3: **Yes: **Diego Serraino, MD, MSc

Reviewer #4: **Yes: **Hedia Bellali, MD, Associate professor in Epidemiology and Public Health

---

## [Author Response · Author response to Decision Letter 2]

23 Jul 2021

all comments have been addressed.

---

## [Decision Letter · Decision Letter 3]

9 Aug 2021

Health extension service utilization and associated factors in east Gojjam zone, North West Ethiopia: community based cross-sectional study

PONE-D-20-30163R3

Dear Dr. Aynalem,

We’re pleased to inform you that your manuscript has been judged scientifically suitable for publication and will be formally accepted for publication once it meets all outstanding technical requirements.

Kind regards,

Ammal Mokhtar Metwally, Ph.D (MD)

Academic Editor

PLOS ONE

Additional Editor Comments (optional):

Reviewers' comments:

Reviewer's Responses to Questions

**Comments to the Author**

1. If the authors have adequately addressed your comments raised in a previous round of review and you feel that this manuscript is now acceptable for publication, you may indicate that here to bypass the “Comments to the Author” section, enter your conflict of interest statement in the “Confidential to Editor” section, and submit your "Accept" recommendation.

Reviewer #3: All comments have been addressed

Reviewer #4: All comments have been addressed

2. Is the manuscript technically sound, and do the data support the conclusions?

Reviewer #3: Yes

Reviewer #4: Yes

3. Has the statistical analysis been performed appropriately and rigorously? 

Reviewer #3: Yes

Reviewer #4: Yes

4. Have the authors made all data underlying the findings in their manuscript fully available?

Reviewer #3: Yes

Reviewer #4: Yes

5. Is the manuscript presented in an intelligible fashion and written in standard English?

Reviewer #3: Yes

Reviewer #4: Yes

6. Review Comments to the Author

Reviewer #3: (No Response)

Reviewer #4: (No Response)

7. PLOS authors have the option to publish the peer review history of their article (what does this mean?). If published, this will include your full peer review and any attached files.

Reviewer #3: No

Reviewer #4: **Yes: **Dr Hedia Bellali, Associate professor in Epidemiology and Public Health, Medical Faculty of Tunis, Tunisia

---

## [Editor Report · Acceptance letter]

11 Aug 2021

PONE-D-20-30163R3 

Health extension service utilization and associated factors in east Gojjam zone, North West Ethiopia: a community-based cross-sectional study 

Dear Dr. Aynalem:

I'm pleased to inform you that your manuscript has been deemed suitable for publication in PLOS ONE. Congratulations! Your manuscript is now with our production department. 

Kind regards, 

on behalf of

Professor Ammal Mokhtar Metwally 

Academic Editor

PLOS ONE